# LOOK, REMEMBER AND REASON: GROUNDED REASONING IN VIDEOS WITH LANGUAGE MODELS

**Apratim Bhattacharyya, Sunny Panchal, Mingu Lee, Reza Pourreza, Pulkit Madan, Roland Memisevic**
Qualcomm AI Research[*]

## ABSTRACT

Multi-modal language models (LM) have recently shown promising performance in high-level reasoning tasks on videos. However, existing methods still fall short in tasks like causal or compositional spatiotemporal reasoning over actions, in which model predictions need to be grounded in fine-grained low-level details, such as object motions and object interactions. In this work, we propose training an LM end-to-end on low-level surrogate tasks, including object detection, re-identification, and tracking, to endow the model with the required low-level visual capabilities. We show that a two-stream video encoder with spatiotemporal attention is effective at capturing the required static and motion-based cues in the video. By leveraging the LM's ability to perform the low-level surrogate tasks, we can cast reasoning in videos as the three-step process of *Look, Remember, Reason*, wherein visual information is extracted using low-level visual skills step-by-step and then integrated to arrive at a final answer. We demonstrate the effectiveness of our framework on diverse visual reasoning tasks from the ACRE, CATER, Something-Else and STAR datasets. Our approach is trainable end-to-end and surpasses state-of-the-art task-specific methods across tasks by a large margin.

## 1 INTRODUCTION

Autoregressive language models (LMs) have shown impressive results on reasoning tasks such as on grade school math problems (Cobbe et al., 2021) and even on LSAT (abs, 2023). Most language models designed for these problems, however, are trained on only textual data. Since many real-world scenarios require reasoning over heterogeneous sensory inputs, *e.g.*, visual cues. multi-modal LMs for images or videos have recently gained traction (Alayrac et al., 2022; Koh et al., 2023; Zhang et al., 2023b; Maaz et al., 2023; Li et al., 2023b;b; Maaz et al., 2023). The focus of current multi-modal LMs for videos have primarily been on high-level question answering and instruction following (Maaz et al., 2023; Li et al., 2023b; Alayrac et al., 2022). However, many visual reasoning problems in videos require grounding in fine-grained low-level information, *i.e.*, recognizing objects, and understanding their spatiotemporal interactions. For example, as shown in Fig. 1, recognition of the high-level compositional action requires a fine-grained understanding of low-level details of motion and interactions between objects. The ability of multi-modal LMs to perform visual reasoning tasks such as compositional action recognition in videos (Materzynska et al., 2020) that require a combination of low-level skills with high-level reasoning has not yet been explored.

To enable multi-modal LMs to solve such reasoning problems we propose our *Look, Remember and Reason (LRR)* multi-modal LM. Our LRR model architecture extracts dense spatiotemporal features from each input frame. This is accomplished using a two-stream attention-based architecture that captures low-level spatial and temporal details (Simonyan & Zisserman, 2014) at each input video frame using top-down cross-attention. Our multi-modal LRR model is grounded to relevant low-level visual information in the scene by stochastically introducing low-level surrogate tasks during training, including object recognition, re-identification, and tracking, at randomly selected time-steps (*c.f.* Fig. 1). We keep these grounded spatiotemporal features, which include low-level visual details, in the working memory, *i.e.*, "remembered" within the context window of the LM. This allows the

---

[*]Qualcomm AI Research is an initiative of Qualcomm Technologies, Inc.

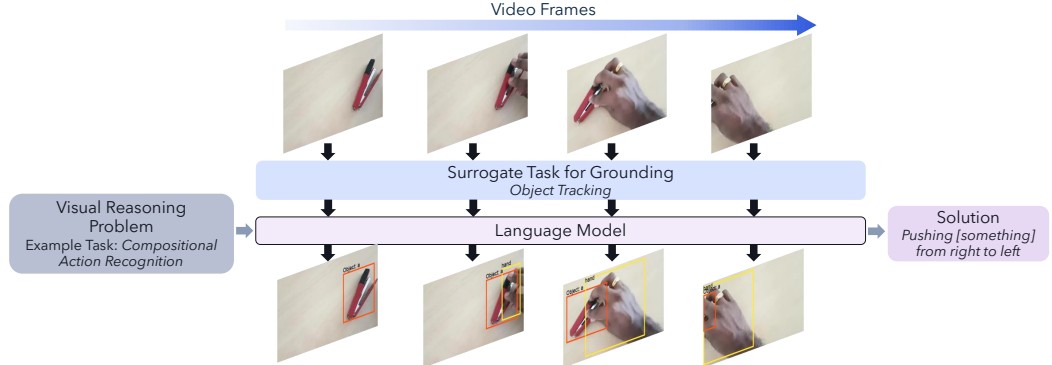

Figure 1. Our Look, Remember, Reason (LRR) model 'looks' at the video frames to extract relevant low-level information, *e.g.*, object motion and interactions, supervised with surrogate tasks like object tracking only during training. It 'remembers' the information from intermediate steps and 'reasons' using the aggregated information.

model to combine the low-level visual features with high-level inferences to "reason" and generate the final responses using our LRR framework as shown in Fig. 1.

Our main contributions are: (i) We highlight the importance of grounding for visual reasoning in multi-modal LMs and propose a novel Look, Remember, and Reason framework to this end to instill the required low-level visual skills in the model using surrogate tasks during training; (ii) We introduce a two-stream video encoder that captures both the scene structure and object motions, crucial for learning the low-level skills; (iii) We demonstrate the effectiveness of our approach on ACRE (Zhang et al., 2021), CATER (Girdhar & Ramanan, 2020), and the real-world Something-Else (Materzynska et al., 2020) and STAR (Wu et al., 2021) datasets. Our approach outperforms the prior state-of-the-art, that is based on highly task-specific architectures, by a large margin—highlighting how our general-purpose LRR model can perform varied and complex spatiotemporal reasoning tasks in videos including causal, compositional and situated reasoning.

## 2  RELATED WORK

**Multi-modal language models.** The use of auto-regressive language models with adapters to process visual inputs is an active area of research. Recently proposed models include Pix2seq (Chen et al., 2022), which is trained to extract low-level visual information from images. Unlike our work, extracting low-level visual information is a goal in itself in that work not a surrogate task, and the model processes images not videos. Other image-based multi-modal LMs include ViperGPT (Surís et al., 2023), VisProg (Gupta & Kembhavi, 2022), and Chameleon (Lu et al., 2023). Similarly, PaLM-E (Driess et al., 2023) provides images and text as interleaved multi-modal latent vectors, allowing it to process multiple images within any part of a sentence which serves as an input to the LM where the model is trained end-to-end. LLaMA-Adapter (Zhang et al., 2023a) introduces an adapter layer to enable multi-modal inputs with the LLaMA model Touvron et al. (2023). FROMAGe (Koh et al., 2023) freezes the language model and fine-tunes the input and output linear layers to encode multi-modal interactions. InstructBLIP (Dai et al., 2023) improves high-level instruction following abilities using instruction fine-tuning. Qwen-VL (Bai et al., 2023) and Kosmos-2 (Peng et al., 2023) propose grounded multi-model LMs that can localize referring expressions in images. In contrast to our work, all these approaches process images not videos.

Video-based multi-modal LMs include Video-ChatGPT (Maaz et al., 2023), VideoChat (Li et al., 2023b), and Valley (Luo et al., 2023). They focus on question answering and instruction following using video. Flamingo (Alayrac et al., 2022), by using a Perceiver as input, is also able to ingest video data, and similarly is trained to infer high-level concepts. In our work, in contrast, we study visual reasoning tasks that require low-level understanding in the form of *e.g.*, motion, actions, and interactions, in addition to high-level reasoning. We show how grounding high-level concepts in these low-level representations greatly improves video-based reasoning tasks.

**Spatiotemporal video grounding.** Recent work on grounding focuses on spatially localizing an object in an image or in a video given a referring expression (Huang et al., 2018; Shi et al., 2019; Vasudevan et al., 2018; Li et al., 2023c). Deng et al. (2021); Yang et al. (2020); Luo et al. (2020); Yang et al. (2019); Kamath et al. (2021) propose end-to-end approaches for the spatial localization in images. For video, Chen et al. (2018; 2019); Zeng et al. (2020); Zhang et al. (2019; 2020);

He et al. (2019); Jang et al. (2023) focus on temporal localization given a natural language query. Methods like STVGBert (Su et al., 2021) and STCAT (Jin et al., 2022) perform spatiotemporal localization of a natural language query using a transformer-based model. Yang et al. (2022) focuses on action localization given a natural language query. Xiao et al. (2023) propose video grounding using Gaussian mask optimization. In contrast to these methods, the approach we introduce performs visual grounding in a pre-trained LM through end-to-end training using surrogate tasks. We show that this makes it possible to instill low-level visual capabilities as required for solving reasoning problems.

**Attention-based models and visual reasoning.** Attention-based models have been studied extensively for visual reasoning Ding et al. (2021); Hu et al. (2017); Hudson & Manning (2018); Kamath et al. (2021); Mahajan & Roth (2020); Santoro et al. (2017). Recent advances include an object-centric encoder and a transformer reasoning module to solve RPM-like benchmarks (Mondal et al., 2023), multi-hop feature modulation (Strub et al., 2018) and cascaded modulation networks (Yao et al., 2018) that use a multi-step comprehension process, neural interpreters (Rahaman et al., 2021) that factorize inference in a self-attention network and ALANS learner (Zhang et al., 2022a) that combines abstract algebra and representation theory. Calibrating concepts and operations Li et al. (2021) enables neural symbolic models to capture underlying data characteristics and perform hierarchical inference. Again, the key difference in our work is that we instill the ability to extract object-centric information within the LM itself, using surrogate tasks, instead of resorting to separate object detection modules.

## 3 LOOK, REMEMBER, REASON

To allow for visual reasoning in videos using language models, we propose a novel *Look, Remember, Reason* framework which grounds the model to the low-level structual and motion information in the visual input required for solving high-level reasoning tasks. Grounding is enabled through low-level visual surrogate tasks, *e.g.* object recognition, tracking and re-identification. In the following, we first describe our auto-regressive LRR architecture including details of our two-stream video encoder, followed by our training pipeline with details of our surrogate tasks.

### 3.1 AUTO-REGRESSIVE PIPELINE

Our LRR model parameterized by $\theta$, as shown in Fig. 2, is based on a pre-trained LM backbone with a two-stream auto-regressive video encoder. Our LRR model receives the visual reasoning problem $\mathbf{Q}$ and, an interleaved sequence $\mathcal{I}$ of video frames $\mathbf{V} = (\mathbf{v}_1, \ldots, \mathbf{v}_{T_v})$ and tokenized text $\mathbf{S} = (\mathbf{s}_1, \ldots, \mathbf{s}_{T_s})$ as input. $\mathbf{S}$ consists of low-level visual surrogate tasks and the answer indicated by <taskname> and <answer> respectively. The input video frame sequence $\mathbf{V}$ is encoded by our two-stream video encoder. The LM backbone receives as input $\mathbf{Q}$, and the interleaved sequence of encoded video frames $\mathbf{V}$ whose positions are indicated with <frame> special tokens and $\mathbf{S}$. Conditioned on $\mathbf{Q}$ and $\mathbf{V}$, we train our LRR model by maximizing log-likelihood of the text sequence $\mathbf{S}$ as,

$$\log\left(p_\theta(\mathbf{S}|\mathbf{Q}, \mathbf{V})\right) = \sum_{t_s} \log\left(p_\theta(\mathbf{s}_{t_s}|\mathbf{s}_1, \ldots, \mathbf{s}_{t_s-1}, \mathbf{v}_1, \ldots, \mathbf{v}_{t_v}, \mathbf{Q})\right) \tag{1}$$

where, $(\mathbf{v}_1, \ldots, \mathbf{v}_{t_v})$ is the interleaved video frame sub-sequence up to the text token $\mathbf{s}_{t_s}$. In Section 3.4, we describe the surrogate tasks included in $\mathbf{S}$. The parameters of the LM backbone are initialized from pre-trained LMs, allowing us to exploit their existing high-level reasoning capabilities. We use LMs from the OPT family (Zhang et al., 2022b), but verified that similar performance can be achieved using other pre-trained models (Gao et al., 2021; Scao et al., 2022). While LM backbones are trained only on text, visual reasoning relies on information from the visual domain $\mathbf{V}$. Therefore, to ground our LRR model, visual information in $\mathbf{V}$ needs to be mapped to the text-based representation space of the LM. The key challenge here is that in contrast to text tokens, videos are highly information dense. To address this, we develop a two-stream video encoder to extract information relevant for the visual reasoning problem at hand, such that responses to the visual reasoning problems are grounded in the fine-grained low-level details of the video.

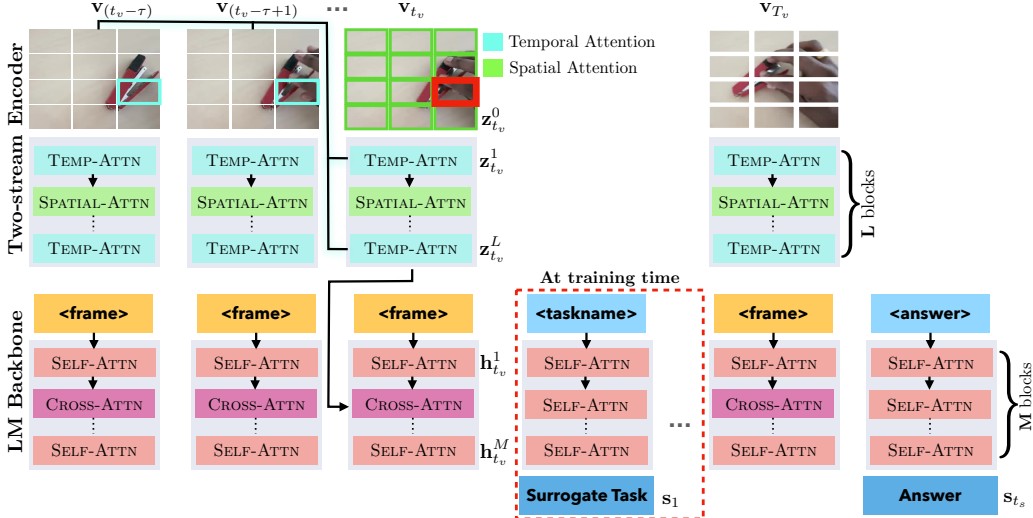

Figure 2. The architecture of our LRR model, highlighting the use of interleaved top-down cross-attention layers in between self-attention layers higher up in the hierarchy.

## 3.2 TWO-STREAM VIDEO ENCODER

Our autoregressive two-stream video encoder exploits divided space-time attention (Bertasius et al., 2021) and generates for each input video frame patch based embeddings that capture relevant low-level spatio-temporal information. Spatial attention captures structural information, *e.g.*, object identities. Temporal attention autoregressively captures object motion and interaction information using the previous $\tau$ frames as a buffer. As a first step, it converts each input video frame $\mathbf{v}_{t_v}$ into $P$ flattened patches $\mathbf{v}_{t_v} = \{\mathbf{v}_{(1,t_v)}, \ldots, \mathbf{v}_{(P,t_v)}\}$ of size $16 \times 16$ (Dosovitskiy et al., 2021) as shown in Fig. 2. Our LRR model applies a linear transformation (LN) on the input patches to generate the initial patch embeddings $\mathbf{z}_{t_v}^0 = \{\mathbf{z}_{(1,t_v)}^0, \ldots, \mathbf{z}_{(P,t_v)}^0\}$ as,

$$\mathbf{z}_{(p,t_v)}^0 = \text{LN}(\mathbf{v}_{(p,t_v)}) + \mathbf{em}_{(p,t_v)} \tag{2}$$

The spatiotemporal positional embedding $\mathbf{em}_{(p,t_v)}$ is added to each patch to aid our video encoder extract spatiotemporal features from each patch in the input video frame. Next, these patch embeddings are processed by the $L$ blocks of our video encoder where at each block spatial or temporal attention is applied. At every block $\ell \in \{1, \ldots, L\}$ of our model, the embeddings are linearly mapped to produce the query, keys, and values required for the space- and time-based attention operations,

$$\mathbf{q}_{(p,t_v)}^\ell = \text{LN}(\mathbf{z}_{(p,t_v)}^\ell), \quad \mathbf{k}_{(p,t_v)}^\ell = \text{LN}(\mathbf{z}_{(p,t_v)}^\ell), \quad \mathbf{v}_{(p,t_v)}^\ell = \text{LN}(\mathbf{z}_{(p,t_v)}^\ell). \tag{3}$$

In case of temporal attention, for the patch embedding $\mathbf{z}_{(p,t_v)}^\ell$ the attention is computed over patch embeddings at the same spatial position $p$ in the $\tau$ previous input video frames, $\mathbf{z}_{(p,(t_v-\tau))}^\ell$ to $\mathbf{z}_{(p,t_v)}^\ell$. For the patch highlighted in red in Fig. 2 the patches where temporal attention is applied in the previous $\tau$ input frames is highlighted in cyan. The temporal attention vector $(\alpha_{\mathcal{T}})_{(p,t_v)}^\ell$ is given by,

$$(\alpha_{\mathcal{T}})_{(p,t_v)}^\ell = \text{SOFTMAX}\left(\frac{\mathbf{q}_{(p,t_v)}^{\ell}{}^\top}{\sqrt{d_m}}[\mathbf{k}_{(p,(t_v-\tau))}^\ell, \ldots, \mathbf{k}_{(p,t_v)}^\ell]\right) \tag{4}$$

where $d_m$ is the dimensionality of the key, queries and values. In contrast, the spatial attention vector, $(\alpha_{\mathcal{S}})_{(p,t_v)}^\ell$, is calculated over all patch embeddings of the current input video frame, highlighted in green in Fig. 2,

$$(\alpha_{\mathcal{S}})_{(p,t_v)}^\ell = \text{SOFTMAX}\left(\frac{\mathbf{q}_{(p,t_v)}^{\ell}{}^\top}{\sqrt{d_m}}[\mathbf{k}_{(1,t_v)}^\ell, \ldots, \mathbf{k}_{(P,t_v)}^\ell]\right) \tag{5}$$

The patch embeddings $\mathbf{z}_{(p,t_v)}^{\ell+1}$ are the weighted sums of the values $\mathbf{v}_{(p,t_v)}^\ell$ computed using the attention weights $(\alpha_{\mathcal{T}})_{(p,t_v)}^\ell$ or $(\alpha_{\mathcal{S}})_{(p,t_v)}^\ell$ in case of temporal or spatial attention, respectively. Finally, the patch embeddings $\mathbf{z}_{t_v}^L$, containing localized information of the scene structure and motion, are obtained as output from the last block $L$ at time-step $t_v$ of the video encoder.

## 3.3 LM Backbone with Top-down Cross Attention

Visual information encoded by the two-stream video encoder is mapped top-down to the LM backbone at positions indicated by the <frame> token at time-steps $t_v$ (Fig. 2) using cross attention layers. This top-down mechanism with cross attention layers, trained using surrogate tasks (Section 3.4), helps us ground the LM backbone to the relevant low-level information from the input video frames.

We modify the backbone LM architecture by inserting cross attention (Cross-Attn) layers (Alayrac et al., 2022) between self-attention (Self-Attn) layers (Fig. 2). The output patch embeddings from our video encoder $\mathbf{z}_{t_v}^L = \{\mathbf{z}_{(1,t_v)}^L, \ldots, \mathbf{z}_{(p,t_v)}^L\}$ are transformed using multi-layer perceptrons (MLP) at every cross-attention layer which help deal with the domain shift between the visual and textual domains. Next, we exploit the rich hierarchical representations in the hidden states $\mathbf{h}_{t_v} = \{\mathbf{h}_{t_v}^1, \ldots, \mathbf{h}_{t_v}^M\}$ of the LM at timestep $t_v$, where $M$ is the number of self-attention layers, to "look" at the visual input and extract visual information in a top-down fashion for grounding. Specifically, we use the representation $\hat{\mathbf{h}}_{t_v}^k$ after the application of the $(k+1)^{\text{th}}$ self-attention layer in the LM. The hidden representation $\hat{\mathbf{h}}_{t_v}^k$ encodes global semantics about the visual reasoning problem $\mathbf{Q}$ and previous video frames upto $t_v$. It serves as the query vector after a linear transformation and the visual features $\mathbf{z}_{t_v}^L$ serves as the keys and values of the cross attention layer respectively,

$$\hat{\mathbf{h}}_{t_v}^k = \text{Self-Attn}(\mathbf{h}_{t_v}^k) \tag{6}$$

$$\hat{\mathbf{z}}_{t_v} = \text{Cross-Attn}(\hat{\mathbf{h}}_{t_v}^k, \mathbf{z}_{t_v}^L) \tag{7}$$

$$\mathbf{h}_{t_v}^{k+1} = \mathbf{h}_{t_v}^k + \hat{\mathbf{h}}_{t_v}^k + \hat{\mathbf{z}}_{t_v} \tag{8}$$

$$\mathbf{h}_{t_v}^{k+1} = \text{FFN}(\mathbf{h}_{t_v}^{k+1}) + \mathbf{h}_{t_v}^{k+1} \tag{9}$$

where FFN is a feedforward layer (Vaswani et al., 2017). As the hidden representation $\mathbf{h}_{t_v}^k$ encodes global semantics, it allows the LM backbone to extract relevant low-level spatiotemporal information in $\hat{\mathbf{z}}_{t_v}$ (Eq. (7)), when trained using our surrogate tasks. This information added to hidden states $\mathbf{h}_{t_v}^{k+1}$ (Eq. (8)) and is implicitly "remembered" within the context window of the LM. This information can be aggregated by the LM to "reason" and arrive at the final answer.

## 3.4 Grounding through Surrogate tasks

To ground our LRR model to the relevant low-level information in the visual input, we utilize surrogate tasks during training. This is illustrated in Fig. 2 where the model is prompted using special tokens (<taskname>) to solve a surrogate task. We consider tasks like object recognition, tracking and re-identification as low-level. From Fig. 1, the ability to recognize the performed action rests upon grounding to the motion and interactions of the hand and the stapler. This makes recognition and tracking constituent low-level capabilities for recognizing the (high-level) compositional action.

Our LRR model is highly flexible and can be prompted to solve a wide range of low-level surrogate tasks. Here, we focus primarily on the tasks of object recognition, localization, re-identification and tracking. These tasks are fundamental to solving a range of visual reasoning problems, and the requisite ground truth can be readily obtained using off-the-shelf vision models (Ren et al., 2017; Tang et al., 2017; Ye et al., 2022). Moreover, these tasks can be encoded as text to be processed by the LM backbone in the following general format: "<taskname> *object id$_1$, object class$_1$, object bounding box$_1$; ... ; object id$_n$, object class$_n$, object bounding box$_n$*", where there are $n$ objects in the scene. The <taskname> special token prompts the model to solve the surrogate task. We use <detect>, <re-identify> and <track> for detection, re-identification, and tracking surrogate tasks respectively. The *object id* is an integer that is assigned by the model based on the spatiotemporal order of appearance and is crucial for re-identification and tracking as same object instance should be assigned the same id across video frames. The bounding box is described as a 4-tuple of the x and y coordinates of the upper left and lower right corners. We provide additional details in the experimental section (Section 4).

Our LRR model is prompted to solve these surrogate tasks at randomly selected time steps during training. Random prompting forgoes the need to include surrogate tasks during inference time, leading to faster inference. Random prompting also benefits training efficiency in the case of long video sequences. To train our LRR model using maximum likelihood as described in Eq. (1). We construct the token sequence $\mathbf{S}$ in $\mathcal{I}$ that includes surrogate tasks for each training example as follows:

Table 1. Evaluation results on the ACRE dataset, where, D.R. – Direct evidence, I.D. – Indirect evidence, S.O. – Screened-off, and B.B. – Backward Blocked subsets (*represents results tested by ourselves.).

| Model | Compositional | | | | | Systematic | | | | |
|---|---|---|---|---|---|---|---|---|---|---|
| | All | D.R. | I.D. | S.O. | B.B. | All | D.R. | I.D. | S.O. | B.B |
| NS-OPT (Zhang et al., 2021) | 69.0 | 92.5 | 76.0 | 88.3 | 13.4 | 67.4 | 94.7 | 88.3 | 82.7 | 16.0 |
| IV-CL† (Sun et al., 2023) | 93.2 | - | - | - | - | 92.6 | - | - | - | - |
| ALOE (Ding et al., 2021) | 91.7 | 97.1 | 90.8 | 96.8 | 78.8 | 93.9 | 97.1 | 71.2 | 98.9 | 94.4 |
| OpenFlamingo* (Awadalla et al., 2023) | 38.2 | 42.6 | 49.5 | 9.9 | 47.6 | 38.6 | 36.5 | 25.8 | 13.7 | 67.6 |
| **LRR (Ours)** | **98.2** | **99.9** | **92.8** | **99.2** | **97.4** | **99.2** | **99.9** | **97.0** | **99.9** | **98.8** |
| LRR (w/o Surrogate tasks) | 38.1 | 38.4 | 30.2 | 26.2 | 50.0 | 36.5 | 35.2 | 28.1 | 20.0 | 55.3 |
| LRR (w/o "Two-stream" encoder) | 92.2 | 98.6 | 77.7 | 96.2 | 85.8 | 92.8 | 98.5 | 76.2 | 96.8 | 89.4 |
| LRR (from scratch) | 87.7 | 96.9 | 57.7 | 91.1 | 84.8 | 88.0 | 96.3 | 58.5 | 91.5 | 86.9 |

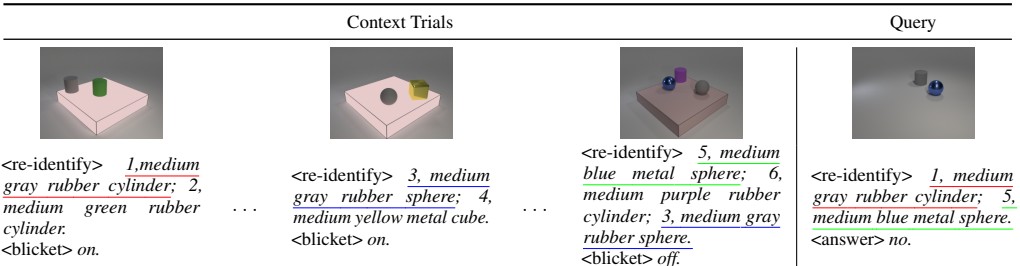

Figure 3. Example solutions to surrogate tasks generated by our LRR model on ACRE. Re-identified objects across context trials are underlined in the same color.

We first randomly select a subset of video frames $(\mathbf{v}_{t_1}, \ldots, \mathbf{v}_{t_k}) \in \mathbf{V}$. We then update the token sequence $\mathbf{S}$ to include the surrogate tasks at these randomly selected time-steps, the beginning of which is marked by a task specific <taskname> special token, *e.g.*, <detect>, <re-identify> and <track>. The surrogate task itself is added in $\mathbf{S}$ as described above.

# 4 EXPERIMENTS

We evaluate on visual reasoning tasks from: ACRE (Zhang et al., 2021), Something-Else Materzynska et al. (2020), CATER (Girdhar & Ramanan, 2020) and STAR (Wu et al., 2021).

**Models and training details.** We fine-tune the pre-trained LM backbone along with the video encoder and cross-attention layers in our LRR model as the visual reasoning problems considered here are challenging and cannot be accurately solved by prompting state of the art LMs such as GPT-4 (Gendron et al., 2023). We focus on the OPT family of LMs Zhang et al. (2022b), particularly OPT-125M/350M/1.3B. We use 4 Nvidia A100 GPUs. Additional details in the appendix.

## 4.1 ACRE

The ACRE dataset (Zhang et al., 2021) evaluates how well vision systems perform causal induction. It focuses on causal discovery through "blicket" detection tests (originally designed for children). A blicket detector activates in the presence of a blicket object. The experiment uses context trials where different objects are placed on the detector, revealing its activation status. Subjects must then determine which objects or combinations would trigger the detector. The ACRE dataset includes 6 context trials and 4 blicket detection tests per sample.

**Surrogate tasks.** The key low-level visual challenge in the ACRE dataset is to associate query objects to the context trials to detect whether the blicket machine is activated. Therefore, we consider the surrogate tasks of object recognition and re-identification. The solution to the surrogate task can be generated by the backbone LM in the following format: "<re-identify> *object id$_1$, object class$_1$; ... ; object id$_n$, object class$_n$* " as shown in Fig. 3. We also found it helpful to introduce an additional surrogate task to identify the state of the blicket machine: "<blicket> *on/off*". These surrogate tasks are introduced randomly during training with a probability of 30% after each context trial or query.

**Baselines and evaluation.** We base our LRR models on the OPT-125M backbone. For comparison (see Table 1), we include the state-of-the-art multi-modal LLM OpenFlamingo (3B-mosaicml/mpt-

Table 2. Evaluation on the Something-Else dataset (*represents results tested by ourselves.).

| Method | Base | | Compositional | |
|---|---|---|---|---|
| | Top-1 | Top-5 | Top-1 | Top-5 |
| STIN + OIE + NL (Materzynska et al., 2020) | 78.1 | 94.5 | 56.2 | 81.3 |
| Video-ChatGPT* (Maaz et al., 2023) | 52.6 | 75.8 | 38.6 | 67.8 |
| LRR (Ours) | **80.2** | **96.1** | **62.0** | **86.3** |
| LRR (w/o Surrogate tasks) | 71.3 | 89.6 | 50.1 | 70.8 |
| LRR (w/o Two-stream encoder) | 73.2 | 90.4 | 53.6 | 76.1 |

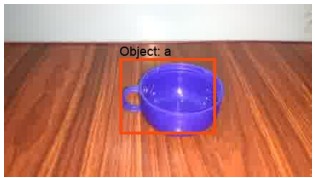 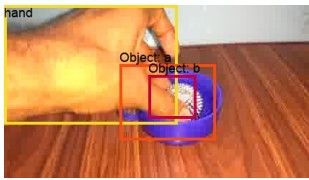 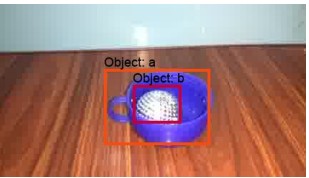

Figure 4. Example solutions to surrogate task tracking generated by our LRR model on Something-Else. Bounding boxes belonging to the same track are highlighted using the same color.

1b-redpajama-200b-dolly; Awadalla et al. (2023)), as it has shown success in reasoning problems involving multiple images and videos. We also test the following ablations to highlight key components: 1. LRR (w/o Surrogate tasks): Tests the importance of the surrogate re-identification task; 2. LRR (w/o Two-stream encoder): Tests the importance of our two-stream encoder's temporal attention mechanism for object re-identification; 3. LRR (trained from scratch): Demonstrates the value of pre-trained LM backbones by using the same OPT-125M architecture without pre-training.

Our LRR model outperforms the state of the art powerful transformer based ALOE (Ding et al., 2021), by a large margin of 6.5% and 5.3% on the compositional and systematic splits respectively. This shows the advantage of our end-to-end LRR model with surrogate tasks over explicit object centric input representations used by ALOE. Similarly, we observe a significant performance gain of 60% over the powerful transformer based multi-modal LLM OpenFlamingo (Awadalla et al., 2023) which highlights the importance of our LRR framework to ground our model to the relevant low-level details – highlighted by the weak performance of the LRR (w/o Surrogate tasks) ablation. We also see that our two-stream encoder improves performance over the single-stream model. The weak performance of the LRR (from scratch) ablation shows that it is crucial to start from a pre-trained LM backbone to exploit its high level reasoning abilities. Finally, although the results in Table 1 use the OPT-125M backbone in our LRR model, we obtain an identical 98.2% and 99.0% accuracy on the compositional and systematic splits respectively with an OPT-1.3B backbone, indicating that our *look, remember, reason* framework is applicable across LM backbone sizes.

## 4.2 SOMETHING-ELSE

This complex, real-world dataset (Materzynska et al., 2020) focuses on compositional action recognition. Building upon the Something-Something dataset (Goyal et al., 2017), it measures compositional generalization by splitting actions into verb, subject, and object combinations. This split allows for benchmarking performance on novel combinations unseen during training, forcing models to develop a nuanced understanding of motion rather than simply associating actions with object types.

**Surrogate tasks.** In this task, we employ tracking as a surrogate task to support the model's ability to capture motion and object interactions. Note that, as object classes are not important for compositional action recognition, the solution to the surrogate task can be generated in the following format: "<track> *object id$_1$, object bounding box$_1$; ... ; object id$_n$, object bounding box$_n$*" as shown in Fig. 4. Surrogate tasks are introduced randomly during training with a probability of 30% after each input video frame.

**Baselines and evaluation.** We base our LRR model on the OPT-125M LM backbone and compare to several baselines and ablations in Table 2. We report results on both the base split and the compositional split with novel action-object combinations. We consider the state-of-the-art STIN + OIE + NL Materzynska et al. (2020) and Video-ChatGPT (Maaz et al., 2023) as baselines. We demonstrate the importance of our surrogate tasks and two-stream video encoder through ablations: LRR (w/o Surrogate Tasks) and (w/o Two-stream encoder).

Table 3. Evaluation on the CATER dataset ([†]results reported only for static camera).

| Method | Static Camera | | | Moving Camera | | |
|---|---|---|---|---|---|---|
| | Top-1($\uparrow$) | Top-5($\uparrow$) | L1(grid;$\downarrow$) | Top-1($\uparrow$) | Top-5($\uparrow$) | L1(grid;$\downarrow$) |
| R3D + NL LSTM Girdhar & Ramanan (2020) | 46.2 | 69.9 | 1.5 | 38.6 | 70.2 | 1.5 |
| ALOE Ding et al. (2021) | 74.0 | 94.0 | 0.44 | 59.7 | 90.1 | 0.69 |
| IV-CL[†] (Sun et al., 2023) | 70.1 | 88.3 | - | - | - | - |
| OPNet[†] (Shamsian et al., 2020) | 74.8 | - | 0.54 | - | - | - |
| Hopper[†] (Zhou et al., 2021) | 73.2 | 93.8 | 0.85 | - | - | - |
| TFC V3D[†] (Zhang, 2022) | 79.7 | 95.5 | 0.47 | - | - | - |
| LRR (Ours) | **84.1** | **97.2** | **0.34** | **80.4** | **96.7** | **0.42** |
| LRR (w/o Surrogate tasks) | 68.5 | 88.7 | 0.65 | 62.7 | 86.7 | 0.77 |
| LRR (w/o Two-stream encoder) | 81.4 | 97.2 | 0.44 | 75.6 | 96.6 | 0.53 |

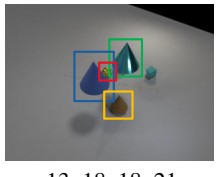 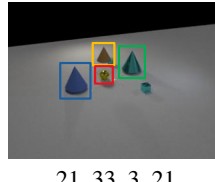 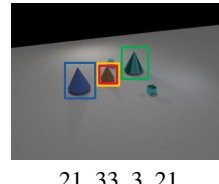 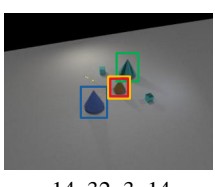

13, 18, 18, 21     21, 33, 3, 21     21, 33, 3, 21     14, 32, 3, 14

Figure 5. Example answers to the tracking surrogate task generated by our LRR model on CATER. Our LRR model is prompted with the "<track>" special token to solve the tracking surrogate task at randomly selected time-steps during training. Object tracks are over the $6 \times 6$ grid on the surface and are highlighted in color.

Our LRR model also outperforms the STIN + OIE + NL baseline by 2.1% and 5.8% Top-1 accuracy on the base and compositional split respectively, which highlights the reasoning ability of grounded LM based architectures. The results also show that the performance of the state of the art multi-modal LM: Video-ChatGPT (finetuned), lags very significantly by 23.4% behind our LRR model. This is because of the CLIP based video encoder in Video-ChatGPT is not well suited for capturing motion features and due to a lack of grounding. The importance of motion features is underscored by the lacking performance of the plain ViT based (w/o Two-stream encoder) ablation: a Top-1 accuracy drop of 8.4% on the compositional split. The importance of grounding is highlighted through our ablation without surrogate tasks: a Top-1 accuracy drop of 11.9% on the compositional split. We illustrate successful tracking of objects in complex real-world scenarios by our LRR model in Fig. 4 and in Appendix B. Note that, although the results in Table 2 uses the OPT-125M backbone, we obtain an identical 61.3% Top-1 and 85.9% Top-5 accuracy on the compositional split with an OPT-1.3B backbone, indicating that our *look, remember, reason* framework is applicable across LM sizes.

## 4.3 CATER

The CATER (Compositional Actions and TEmporal Reasoning) dataset challenges models to recognize complex object movement patterns requiring long-term reasoning. The most difficult task is adversarial target tracking, where a "snitch" object must be located at the end of a video sequence despite potential occlusion or containment within other objects. This is formulated as a classification problem on a $6 \times 6$ grid. The dataset includes static and moving camera splits, with the latter posing additional challenges due to the need for long-term spatiotemporal analysis.

**Surrogate tasks.** We employ (multi-target) tracking as a surrogate task for which the solution can be expressed in the following format: "<track> *object id$_1$, object grid position$_1$; ... ; object id$_n$, object grid position$_n$*" as shown in Fig. 5. The tracking task includes the medium and large cones in the scene, as these objects can occlude the snitch. We use grid positions instead of bounding boxes (unlike Something-Else) because the final goal is to predict the grid position of the snitch. Surrogate tasks are introduced randomly during training with a probability of 30% after each input frame.

**Baselines and evaluation.** Our model uses the OPT-125M backbone (results in Table 3). We ablate both the surrogate (multi-target) tracking task (w/o Surrogate tasks) and our Two-stream video encoder (w/o Two-stream Encoder) to demonstrate their importance. Following ALOE Ding et al. (2021), we jointly train on static and moving camera splits.

The state-of-the-art powerful transformer-based IV-CL (Sun et al., 2023), OPNet (Shamsian et al., 2020), Hopper (Zhou et al., 2021), TFC V3D Depthwise (Zhang, 2022) and Loci (Traub et al., 2023) report results only on the static camera split. Loci (Traub et al., 2023) reports an impressive 90.7%

Table 4. Evaluation of our LRR model on STAR (validation set).

| Method | Int. | Seq. | Pre. | Fea. | Overall ↑ |
|---|---|---|---|---|---|
| Internvideo (Wang et al., 2022) | 62.7 | 65.6 | 54.9 | 51.9 | 58.7 |
| BLIP-2 (Li et al., 2023a) | 65.4 | 69.0 | 59.7 | 54.2 | 62.0 |
| SeViLA (Yu et al., 2023) | 63.7 | 70.4 | 63.1 | 62.4 | 64.9 |
| LRR (Ours) | **73.7** | **71.0** | **71.3** | **65.1** | **70.5** |
| LRR (w/o Surrogate tasks) | 54.5 | 48.7 | 44.3 | 45.5 | 48.2 |

accuracy on the static camera split, but it is not applicable to the moving camera split due to its static background and camera model. Our LRR model outperforms TFC V3D Depthwise (Zhang, 2022) model on the static camera by 4.4% and ALOE (Ding et al., 2021) on the challenging moving camera split by 20.7%. The large performance gain over the LRR (w/o Surrogate tasks) baseline shows the advantage of surrogate tracking tasks, without which the model is not grounded to the motion of the cones and hence fails in cases where the snitch is contained by the cones. Qualitative examples in Fig. 5 illustrates that our model is able to successfully track objects in cases of recursive containment and is robust to moving cameras. Finally, the performance advantage over the LRR (w/o Two-stream encoder) confirms that our LRR model is able to better capture the motion of the objects.

## 4.4   STAR

STAR (Wu et al., 2021) is a situated spatio-temporal reasoning benchmark, consisting of questions built upon real-world videos associated with human actions and interactions. The STAR benchmark thus requires an understanding of low-level human motion, actions and object interactions. However, the STAR dataset does not contain dense object annotations, *e.g*., object tracks, unlike the CATER and Something-Else datasets. It only contains sparse spatio-temporal scene graphs which cover selected keyframes. We introduce object recognition surrogate tasks to localize objects on these keyframes based on the scene graphs. Furthermore, we jointly train our LRR model to recognize actions from the Kinetics (Kay et al., 2017) and Moments in Time (Monfort et al., 2020) dataset, as well as on surrogate tracking tasks from the Something-Else dataset (see Section 4.2) to improve the understanding of object interactions. We also regularize on textual data.

Our LRR model with an OPT-350M LM backbone achieves 70.5% overall accuracy and significantly outperforms powerful trasnformer-based state of the art models, such as SeViLA (Yu et al., 2023), Internvideo (Wang et al., 2022) and BLIP-2 (Li et al., 2023a) as shown in Table 4. Following SeViLA (Yu et al., 2023), we report results on the validation set. Note that methods such as SeViLA are trained on a much larger set of image/video - text pairs and, use the 4.1B parameter BLIP as a video encoder and the 3B parameter Flan-T5 XL as an language model (*c.f*. Section 4.4 in Yu et al. (2023)). In contrast, our LRR model contains fewer parameters and is trained on a much smaller training set. Curicially, our training set includes carefully selected surrogate tasks to endow the model with the requisite low-level visual capabilities. To illustrate this, we include a ablation of our LRR model without any surrogate tasks (w/o Surrogate tasks).

In addition to the results on the STAR validation set presented above we also evaluate our LRR model on the STAR challenge leaderboard. The results can be found: here (ranked 1[st] as of January 2024.)

## 5   CONCLUSION

We show that off-the-shelf LMs can solve complex visual reasoning tasks on videos using our LRR framework. In this framework, LMs equipped with a two-stream video encoder are supervised and grounded using surrogate tasks. Grounding ensures that the LM can utilize relevant low-level visual cues from in the input video to make predictions. Grounding predictions to low-level visual cues combined with the high-level reasoning ability of the LM is the key to the success of the model. Our LRR model outperforms the state of the art by 6.5% and 5.3% on the compositional and systematic splits of the ACRE dataset; by 5.8% Top-1 accuracy on the compositional split of the Something-Else dataset; by 4.4% Top-1 accuracy on static camera and 20.7% Top-1 accuracy on moving camera splits of the CATER dataset; and by 5.6% overall accuracy on STAR.

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

APPENDIX

## A OVERVIEW

Here we provide, 1. Results on a "generalist" LRR model. 2. Surrogate tasks and overfitting. 3. Additional training details including hyper-parameters used for evaluation in Section 4 of the main paper across ACRE, Something-Else and CATER datasets. 4. Additional experiments to compare our LRR approach to "Chain-of-thought" (Wei et al., 2022) based approaches. 5. Additional qualitative examples across ACRE, Something-Else and CATER datasets.

## B GENERALIST MODEL

The results in the main paper show that we can leverage the reasoning capabilities of pre-trained LMs to create "specialist" models with state of the art performance on a variety of visual reasoning tasks using end-to-end learning. Next, as both the input modality and the acquired visual skills are generic and in principle applicable across tasks, we train the model with OPT-125M backbone jointly on the two video-based reasoning tasks discussed above, CATER and Something-Else.[1] This is a step towards developing a grounded "generalist" for performing reasoning in videos. Despite commonalities in underlying low-level features, the diverse nature of reasoning objectives in the two tasks makes this highly challenging. As shown in Table 5, our LRR (Joint) model outperforms even the dataset specific fine-tuned state of the art models (SOTA) (Materzynska et al., 2020; Zhang, 2022; Ding et al., 2021) on these highly diverse visual reasoning tasks.

Table 5. Evaluation of our LRR model trained jointly on CATER and Something-Else.

| Method | Something-Else | | CATER: Static Camera | | CATER: Moving Camera | |
|---|---|---|---|---|---|---|
| | Top 1↑ | Top 5↑ | Top 1↑ | Top 5↑ | Top 1↑ | Top 5↑ |
| SOTA (*c.f*. Tables 2 and 3) | 56.2 | 81.3 | 79.7 | 95.5 | 59.7 | 90.1 |
| LRR (Joint, Ours) | **61.1** | **85.4** | **80.6** | **97.2** | **73.7** | **95.6** |

## C SURROGATE TASKS AND OVERFITTING

To highlight that our surrogate tasks prevent our LRR model from overfitting, we report training and test accuracy of our LRR model trained with and without surrogate tasks on the compositional split of Something Else and on the moving camera split of CATER in Table 6.

The results show that our surrogate tasks clearly prevent overfitting as the model is grounded to the fine-grained low-level details and is thus able to better "understand" the task at hand.

## D ACRE

**Additional training details.** We trained our LRR model with the OPT-125M and OPT-1.3B backbone until convergence ($\sim$ 500k iterations) with a batch size of 4. We use the AdamW optimizer (Loshchilov & Hutter, 2019) with a learning rate of $1 \times 10^{-5}$, $\beta_1 = 0.9$, $\beta_2 = 0.95$ and $\lambda$ (weight decay) = 0.1 and gradient clipping with a norm of 1.0.

**Comparsion to "Chain-of-thought".** As described in Section 3.4 of the main paper, we prompt our LRR model to solve certain surrogate tasks at randomly selected time steps. It is also possible to include surrogate tasks after every time-step. In this case, this would resemble a "Chain of Thought" Wei et al. (2022) like process, where the final answer would depend upon the surrogate tasks solved at inference time. However, this is very inefficient and thus impractical, especially for long video sequences. We compare both of these approaches in Table 7. LRR (Random 30%, Ours) where we prompt our LRR model with the OPT-125M backbone with a probability of 30% an input video frame to solve surrogate tasks *only* during training. LRR (Every frame) where the model is prompted to solve surrogate tasks after every frame during training and inference. Note, that in case of the LRR (Every frame) model it is necessary to solve surrogate tasks during inference as the (autoregressive LM) model always sees such tasks during training. We see from the results in Table 7, that the

---

[1]It may be possible to include datasets like ACRE by treating images as still videos, but we leave this for future work.

Table 6. Results highlighting that our LRR model prevents overfitting.

| Method | Something-Else (Top 1 ↑) | | CATER: Moving Camera (Top 1 ↑) | |
|---|---|---|---|---|
| | Train | Test | Train | Test |
| LRR (w/o Surrogate tasks) | 92.6 | 50.1 | 99.4 | 62.7 |
| LRR (Ours) | 87.3 | 62.0 | 89.7 | 80.4 |

Table 7. Evaluation of our random prompting strategy on ACRE *c.f.* Table 2 in the main paper.

| Method | Compositional ↑ | Systematic ↑ | Inference Speed (msec) ↑ |
|---|---|---|---|
| ALOE (Ding et al., 2021) | 91.7 | 93.9 | - |
| LRR (Random 30% during training, Ours) | 98.2 | 99.2 | **61** |
| LRR (Every frame) | **99.3** | **99.5** | 1415 |

performance of LRR (Random 30% during training, Ours) is comparable with LRR (Every frame) while the inference speed is order of magnitudes faster. We report the inference speed in milliseconds on a single Nvidia A100 GPU. This is because our random prompting during training distills the relevant low-level information into the hidden states of the LM backbone at a feature level and thus does not require solving surrogate tasks at inference time in a COT like fashion.

**Additional qualitative examples.** We include additional qualitative examples in Fig. 6 highlighting surrogate tasks. These examples illustrate that using our surrogate re-identification task, our LRR model can aggregate information from multiple context trials to arrive at the final answer – following the paradigm of "Look, Remember, Reason".

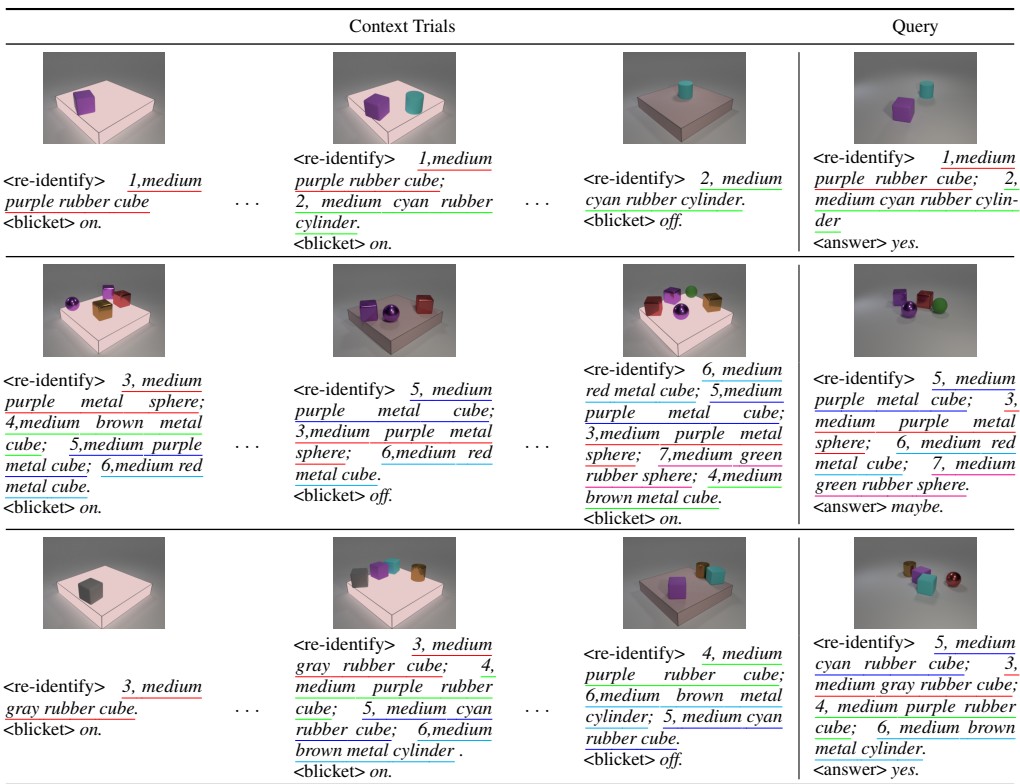

Figure 6. Example solutions to surrogate tasks generated by our LRR model on ACRE. Re-identified objects across context trials are underlined in the same color.

## E  SOMETHING-ELSE

**Additional training details.** We trained our LRR model with the OPT-125M and OPT-1.3B backbone until convergence ($\sim$ 700k iterations) with a batch size of 4. We use the AdamW optimizer (Loshchilov & Hutter, 2019) with a learning rate of $1 \times 10^{-5}$, $\beta_1 = 0.9$, $\beta_2 = 0.95$ and $\lambda$ (weight decay) $= 0.1$ and gradient clipping with a norm of $1.0$. We also employ random data augmentation using RandAugment (Cubuk et al., 2020) with a magnitude of 15 and label smoothing with $\epsilon = 0.3$, where applicable (for fairness).

**Additional qualitative examples.** The qualitative examples in Fig. 4 show that our LRR model can deal with visually complex real-world scenarios such as severe occlusions, *e.g.*, the silver ball in Fig. 4 (bottom row) is occluded by the hand. Here in Fig. 7, we include additional qualitative examples further highlighting that our LRR model can deal with visually complex real-world scenarios such as severe deformations, *e.g.*, the piece of paper in Fig. 7 second row which is torn into two pieces; severe changes in appearance, *e.g.*, the pen in Fig. 7 third row, the teal case in the fifth row; motion blur in Fig. 7 last row.

Finally, these examples illustrate that using our surrogate tracking task, our LRR model can aggregate information from multiple context trials to arrive at the final answer – following the paradigm of "Look, Remember, Reason".

## F  CATER

**Additional training details.** We trained our LRR model with the OPT-125M backbone until convergence ($\sim$ 600k iterations) with a batch size of 4. We use the AdamW optimizer (Loshchilov & Hutter, 2019) with a learning rate of $1 \times 10^{-5}$, $\beta_1 = 0.9$, $\beta_2 = 0.95$ and $\lambda$ (weight decay) $= 0.1$. We use gradient clipping with a norm of $1.0$.

**Additional qualitative examples.** We include additional qualitative examples including both the static and moving camera splits in Fig. 8, highlighting the surrogate tracking task. We see that our LRR model can successfully deal with containment in both static camera (rows 1-3; Fig. 8) and moving camera (row 4-5; Fig. 8) settings, due to our rationale that explicitly tracks cones and the snitch. Note that the example in row 5 in Fig. 8 from the moving camera split is especially challenging due to recursive containment.

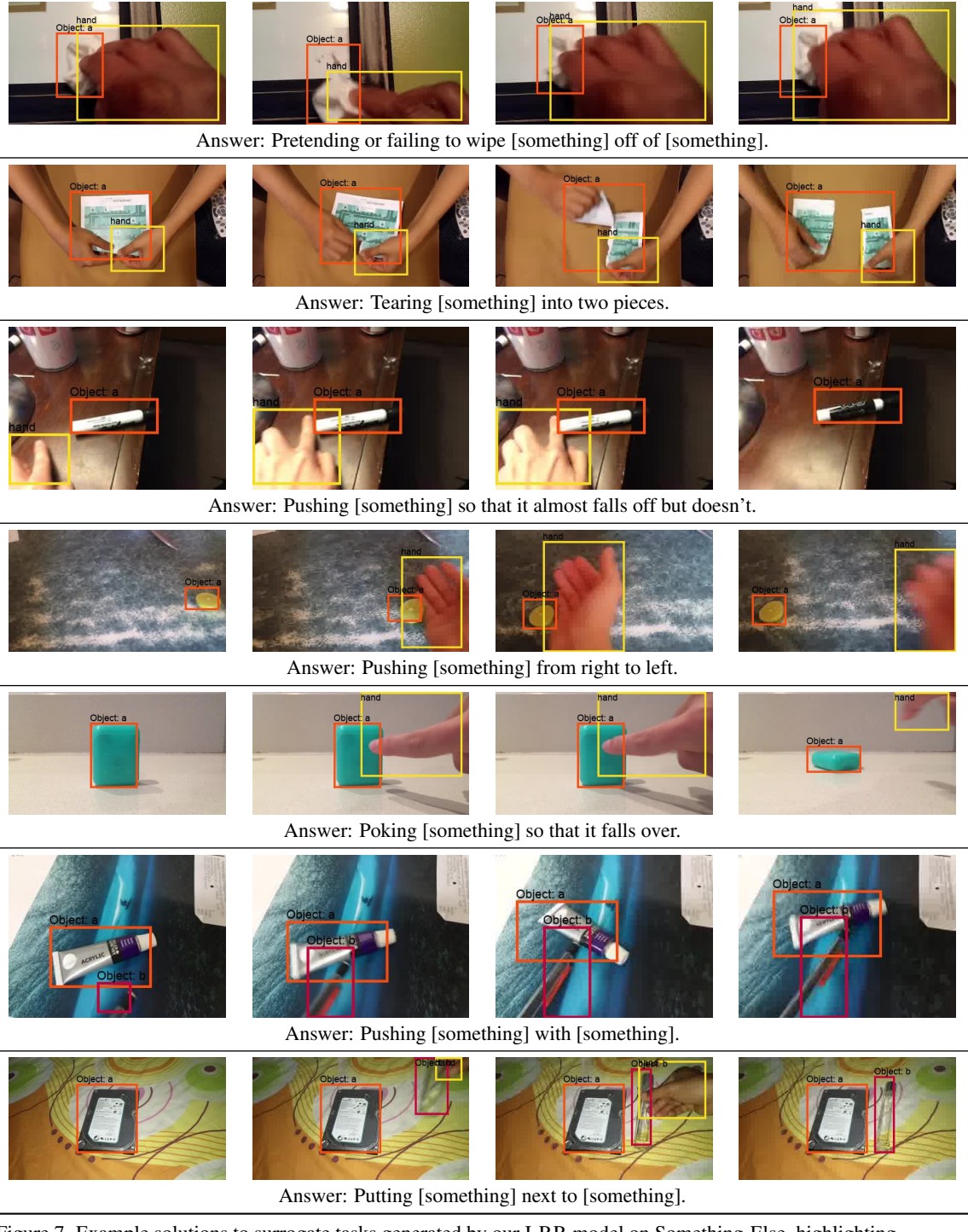

Figure 7. Example solutions to surrogate tasks generated by our LRR model on Something-Else, highlighting object tracks and bounding boxes

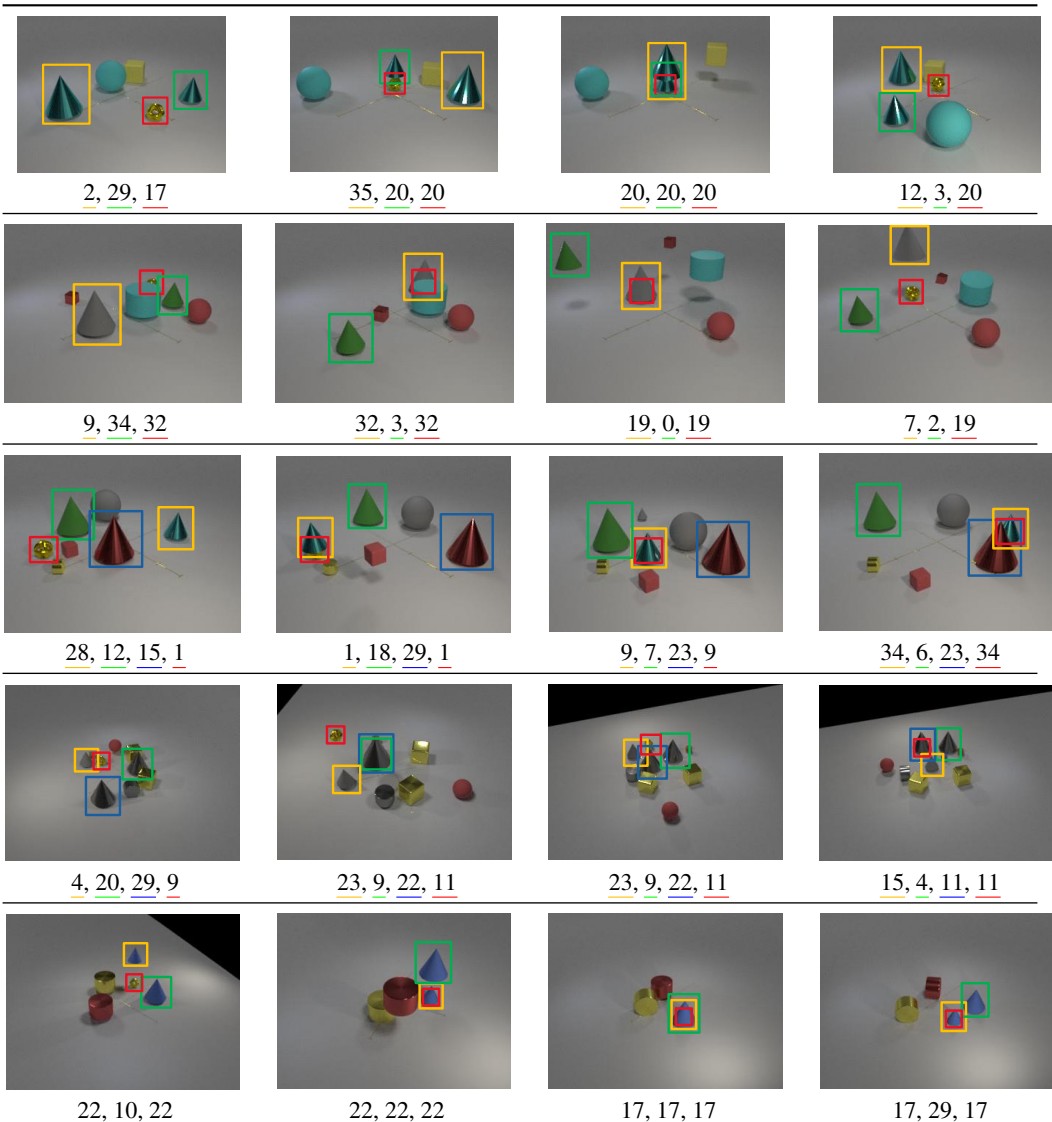

Figure 8. Example solution to the tracking surrogate tasks generated by our LRR model on CATER. We show the predicted grid locations of the cones and the snitch below.

