# OpenReview forum: "Look, Remember and Reason: Grounded Reasoning in Videos with Language Models"
_ICLR.cc/2024/Conference — ICLR 2024 poster_

### Official Review · Reviewer_Fa1W · 2023-10-30

**Soundness:** 3 good
**Presentation:** 3 good
**Contribution:** 3 good
**Rating:** 6
**Confidence:** 5

**Summary:**

The authors claim that they propose training an LM end-to-end on low-level surrogate tasks, including object detection, re-identification, and tracking, to endow the model with the required low-level visual capabilities.

**Strengths:**

The authors claim that they propose training an LM end-to-end on low-level surrogate tasks, including object detection, re-identification, and tracking, to endow the model with the required low-level visual capabilities.

**Weaknesses:**

1. In the experiments, the authors primarily focus on conducting investigations using synthetic datasets, particularly the ACRE dataset. However, it raises concerns about the generalizability of the conclusions/findings obtained from synthetic datasets to real-world datasets.

2. The experimental results primarily focus on classical models. However, the generalizability of the conclusions/findings derived from these classical models to more powerful transformer-based models, such as the models mentioned in *Related Work* part, remains a concern.

**Questions:**

Please refer to Weakness.

---

> ### Author Response · Authors · 2023-11-16
> **Response to Reviewer Fa1W**
>
> We thank the reviewer for their helpful comments and for recognizing that our proposed surrogate tasks such as object detection, re-identification, and tracking, endows our LRR model with the required low-level visual capabilities.
>
> **Synthetic datasets:**  We would like to emphasize that recent work has shown that even on synthetic datasets such as ACRE, state of the art LLMs such as GPT-4 do not perform well (Gendron et. al. 2023). Moreover, we include results on the challenging real-world Something-Else dataset in Section 4.2. We additionally include results on the challenging real-world STAR (Wu et. al. 2021) dataset below and in Appendix B in the paper.
>
> Specifically, note that the results in Fig. 4 and Fig. 7 (Appendix) on Something-Else clearly show that our LRR model can deal with visually complex real-world scenarios. Our LRR model can deal with: severe occlusions, e.g., the silver ball in Fig. 4 bottom row is occluded by the hand; severe deformations, e.g., the piece of paper in Fig. 7 second row which is torn into two pieces; severe changes in appearance, e.g., the pen in Fig. 7 third row. It outperforms the state of the art STIN + OIE + NL model and the recently proposed Video-ChatGPT multi-modal LLM by a significant margin.
>
> We additionally include results on the visually complex STAR dataset in Appendix B. The STAR benchmark (https://bobbywu.com/STAR/) consists of questions built upon real-world videos associated with human actions and interactions. Our LRR model with an OPT-350M LM backbone **achieves 66.7% accuracy** and outperforms prior state of the art models such as SeViLA (Yu et. al., 2023), Internvideo (Wang et.al. 2022) and BLIP-2 (Li et. al. 2023). Further details can be found in Appendix B, where we also include ablations of our LRR model (w/o Surrogate tasks) which highlights the advantage of using surrogate tasks during training.
>
> | Method | Int. | Seq. | Pre. | Fea. | Overall|
> | -------- | :-------: | :-------: | :-------: | :-------: | :-------: |
> | Internvideo | 62.7 | 65.6 | 54.9 | 51.9 | 58.7 |
> | BLIP-2 | 65.4 | 69.0 | 59.7 | 54.2 | 62.0 |
> | SeViLA | 63.7 | 70.4 | 63.1 | **62.4** | 64.9 |
> | LRR (ours) | **79.6** | **71.3** | **63.2** | 52.8 | **66.7** |
>
> Note that methods such as SeViLA are trained on a much larger set of image/video - text pairs and, use the 4.1B parameter BLIP as a video encoder and the 3B parameter Flan-T5 XL as an language model (c.f. Section 4.4 in Yu et al. (2023)). In contrast, our LRR model is trained on a much smaller dataset but one that includes low-level surrogate tasks to endow the model with the requisite low-level visual capabilities.
>
> Finally, in terms of visual complexity, CATER (especially the moving camera split) is complex and challenging as shown in Fig. 5. The results in Fig. 5 and Table 3 indicate that our LRR model can not only detect occlusions for tracking the snitch, but also understand camera motion based just on visual observations, e.g., motion of the background.
> To conclude, the results on Something-Else, STAR, CATER and ACRE clearly show that our LRR model can deal with highly varied complex real-world visual inputs and reasoning tasks.
>
> **Classical models:** We would like to emphasize that we include powerful transformer-based models in our evaluation. In Table 1, we include the recently proposed state of the art OpenFlamingo (Awadalla et al., 2023) multimodal LLM as a baseline. Furthermore, IV-CL (Sun et al., 2023) and ALOE (Ding et. al., 2021) in Table 1 are also large transformer based models. In Table 2, we include the recently proposed state of the art Video-ChatGPT (Maaz et al., 2023) multimodal LLM as a baseline. In Table 3, we compare to the large transformer based IV-CL (Sun et al., 2023) and ALOE (Ding et. al., 2021) models. In Table 5 (Appendix), we compare to SeViLA (Yu et. al., 2023), Internvideo (Wang et.al. 2022) and BLIP-2 (Li et. al. 2023), all of which are large multi-modal transformer based models. Note that, we are first to propose the use of language models for solving complex reasoning tasks such as causal and spatiotemporal compositional reasoning, while ensuring that we always compare to state of art models across all tasks. We will clarify further in the final version.
>
> **References**
>
> [1] STAR: A Benchmark for Situated Reasoning in Real-World Videos, Wu et. al. NeurIPS 2021.
>
> [2] Self-Chained Image-Language Model for Video Localization and Question Answering, Yu et. al. arXiv 2023.
>
> [3] Blip-2: Bootstrapping language-image pre-training with frozen image encoders and large language models, Li et. al. arXiv 2023.
>
> [4] Internvideo: General video foundation models via generative and discriminative learning, Wang et. al., arXiv 2023.
>
> [5] Large Language Models Are Not Abstract Reasoners, Gendron et. al. arXiv 2023.

---

> > ### Author Response · Authors · 2023-11-20
> > **Evaluation Datasets**
> >
> > **STAR Competition Leaderboard** In addition to the results on the STAR validation set presented above (following the protocol of SeViLA (Yu et. al., 2023)), additionally we also evaluate our LRR model on the STAR challenge leaderboard. The results can be found here: https://eval.ai/web/challenges/challenge-page/1325/leaderboard/3328/Mean, where our LRR model achieves 65.75% overall accuracy and currently **ranked 1st** (as of 20th Nov 2023).
> >
> > This again illustrates the capabilities of our LRR model on *visually complex real-world datasets*.

---

> > ### Comment · Reviewer_Fa1W · 2023-11-22
> >
> > The author has answered my question, but I don't find the relevant sentence in the revised version. Therefore my rating is raised from 5 to 6.

---

> ### Author Response · Authors · 2023-11-22
> **Response regarding revised version**
>
> Thank you for raising your score. We are happy to have answered your questions and for helping us improve our manuscript further.
>
> **Revised Manuscript** We have made further revisions to the manuscript, where we discuss complex real-world scenes from the Something-Else dataset in Section 4.2 and Appendix E. We have also updated Appendix B with the results of the (complex real-world) STAR competition (where we are currently **ranked 1st** on the official leaderboard).
>
> We have updated Sections 4.1, 4.2, 4.3 and Appendix B to better highlight comparison to state of the art powerful transformer based models including state of the art multi-modal LLMs, e.g., OpenFlamingo (Awadalla et al., 2023), Video-ChatGPT (Maaz et al., 2023), IV-CL (Sun et al., 2023), ALOE (Ding et. al., 2021), SeViLA (Yu et. al., 2023), Internvideo (Wang et.al. 2022), BLIP-2 (Li et. al. 2023) among others.
>
> All revisions to the manuscript are highlighted in blue.
>
> We would be glad if you could please point out the specific relevant sentence that cannot be found in the revised text. We look forward to your reply.

---

### Official Review · Reviewer_4932 · 2023-10-31

**Soundness:** 3 good
**Presentation:** 3 good
**Contribution:** 3 good
**Rating:** 6
**Confidence:** 3

**Summary:**

Thank you for submitting your manuscript. The proposed three-step process of Look, Remember, Reason for training a Language Model (LM) end-to-end on low-level surrogate tasks, which include object detection, re-identification, and tracking, is indeed novel.

**Strengths:**

The approach presented is intriguing and demonstrates significant performance improvements.

**Weaknesses:**

1-Throughout the paper, there's a recurring mention of "low-level surrogate tasks". Could the authors elucidate the definition of these low-level tasks? Moreover, how do they differ from high-level tasks?

2-The Look, Remember, Reason (LRR) model framework is innovative. However, there seems to be a gap in explicitly correlating this framework with the actual operations carried out in the method. The unique contributions of the "Remember" and "Reason" steps, in particular, are not clearly highlighted. It would be beneficial for the readers if the authors can provide a clearer mapping of these steps to their corresponding operations.

3-Will the codebase for the presented method be made publicly available?

4-Regarding the results of Video-ChatGPT on the Something-Else dataset: Were these results replicated by the authors? I couldn't find a direct reference to such results in the original Video-ChatGPT paper.

**Questions:**

see Weaknesses

---

> ### Author Response · Authors · 2023-11-16
> **Response to Reviewer 4932**
>
> We thank the reviewer for the helpful comments and for recognizing that our Look, Remember, Reason (LRR) model framework is innovative. We also thank the reviewer for recognizing that our LRR framework is intriguing and demonstrates significant performance improvements.
>
> **Low-level surrogate tasks:** We consider tasks like object recognition, tracking and re-identification as low-level tasks. These are computer vision tasks which play a crucial role in solving high-level reasoning tasks, like those found in ACRE, CATER, Something-Else and STAR. For example, in the case of Something-Else, the ability to recognize the performed action in Fig. 4 (top) rests upon grounding our LRR model to the motion and interaction of the hand and the stapler. This makes object recognition and tracking constituent low-level capabilities for recognizing the (high-level) compositional action. We have updated the text in Section 3.4 to make this clearer. We would be glad to receive further feedback to help improve the text further for the final version.
>
> **"Remember" and "Reason" steps:** We apologize for any issues with clarity. As mentioned in Section 3.3, in the “remember” step our LRR model remembers low-level visual information by storing it within the context window of the LLM backbone. In the “reason” step, our LRR model aggregates this low-level visual information to arrive at the final answer (see also Fig. 1). This aggregation happens implicitly through the attention mechanism of the backbone LLM model. We have improved the clarity of the text in Section 1, Section 3.3 and 3.4 in the updated pdf. The updated text is in blue. We would be glad to receive additional feedback to help improve the text further for the final version.
>
> **Codebase:** Code will be released, pending legal review.
>
> **Video-ChatGPT:** The Video-ChatGPT model was tested by ourselves. We have updated Table 2 to indicate this. We use the publicly available codebase (https://github.com/mbzuai-oryx/Video-ChatGPT). We obtained the best results by finetuning the Video-ChatGPT model based on the information in the publicly available codebase and the paper (see Maaz et. al. 2023). We have updated the text in Section 4.2 (in blue).
>
> **References**
>
> [1] Video-ChatGPT: Towards Detailed Video Understanding via Large Vision and Language Models, Maaz et. al. 2023.

---

### Official Review · Reviewer_5Fo1 · 2023-11-04

**Soundness:** 3 good
**Presentation:** 3 good
**Contribution:** 3 good
**Rating:** 8
**Confidence:** 3

**Summary:**

The paper proposes a Look, Remember, Reason (LRR) framework to enable language models to perform visual reasoning in videos. The proposed LRR framework uses a two-stream video encoder to extract dense spatiotemporal features from video frames capturing structural and motion cues. The language model backbone has cross-attention layers inserted between its self-attention layers to enable top-down attention over the visual features. This allows the model to extract relevant visual information based on the reasoning task. LRR is trained end-to-end using surrogate tasks like object detection, re-identification and tracking. These provide supervision to teach the model the required low-level visual skills.

The authors also demonstrate that training LRR jointly on multiple datasets leads to a "generalist" model that performs competitively compared to task-specific "specialist" models. In the experimental results, the authors demonstrate that the LRR models significantly outperform prior state-of-the-art on challenging visual reasoning tasks from the ACRE, Something-Else, and CATER datasets, showing the benefit of the proposed grounded reasoning approach.

**Strengths:**

Demonstrates strong performance on multiple challenging visual reasoning datasets by grounding the language model in low-level visual details.
+ Good demonstration of using surrogate tasks and end to end training

**Weaknesses:**

- The datasets used are rather simple with low visual complexity, such as CATER.

**Questions:**

1) Could you comment on the nature of surrogate tasks? Are there some tasks that are more suited for reasoning vs others. Do low level recognition tasks (choice in the paper) work better.
2) Is there evidence that LRR is not ovefitting to these simplistic datasets due to surrogate tasks?

---

> ### Author Response · Authors · 2023-11-16
> **Response to Reviewer 5Fo1 [1/2]**
>
> We thank the reviewer for the helpful comments and for recognizing that our LRR model demonstrates strong performance on multiple challenging visual reasoning datasets by grounding the language model in low-level visual details.
>
> **Low visual complexity:** We would like to emphasize that recent work has shown that even on synthetic datasets such as ACRE, state of the art LLMs such as GPT-4 do not perform well (Gendron et. al. 2023). Moreover, note that we include results on the challenging real-world Something-Else dataset in Section 4.2. We additionally include results on the challenging STAR (Wu et. al. 2021) dataset below and in Appendix B in the paper.
>
> Specifically, note that the results in Fig. 4 and Fig. 7 (Appendix) on Something-Else clearly show that our LRR model can deal with visually complex real-world scenarios. Our LRR model can deal with: severe occlusions, e.g., the silver ball in Fig. 4 bottom row is occluded by the hand; severe deformations, e.g., the piece of paper in Fig. 7 second row which is torn into two pieces; severe changes in appearance, e.g., the pen in Fig. 7 third row. It outperforms the state of the art STIN + OIE + NL model and the recently proposed Video-ChatGPT multi-modal LLM by a significant margin.
>
> We additionally include results on the visually complex STAR dataset in Appendix B. The STAR benchmark (https://bobbywu.com/STAR/) consists of questions built upon real-world videos associated with human actions and interactions. Our LRR model with an OPT-350M LM backbone **achieves 66.7% accuracy** and outperforms prior state of the art models such as SeViLA (Yu et. al., 2023), Internvideo (Wang et.al. 2022) and BLIP-2 (Li et. al. 2023). Further details can be found in Appendix B, where we also include ablations of our LRR model (w/o Surrogate tasks) which highlights the advantage of using surrogate tasks during training.
>
> | Method | Int. | Seq. | Pre. | Fea. | Overall|
> | -------- | :-------: | :-------: | :-------: | :-------: | :-------: |
> | Internvideo | 62.7 | 65.6 | 54.9 | 51.9 | 58.7 |
> | BLIP-2 | 65.4 | 69.0 | 59.7 | 54.2 | 62.0 |
> | SeViLA | 63.7 | 70.4 | 63.1 | **62.4** | 64.9 |
> | LRR (ours) | **79.6** | **71.3** | **63.2** | 52.8 | **66.7** |
>
> Note that methods such as SeViLA are trained on a much larger set of image/video - text pairs and, use the 4.1B parameter BLIP as a video encoder and the 3B parameter Flan-T5 XL as an language model (c.f. Section 4.4 in Yu et al. (2023)). In contrast, our LRR model is trained on a much smaller dataset but one that includes low-level surrogate tasks to endow the model with the requisite low-level visual capabilities.
>
> Finally, we would like to emphasize that CATER, especially the moving camera split, is visually complex and challenging as shown in Fig. 5. The results in Fig. 5 and Table 3 indicate that our LRR model can not only detect occlusions for tracking the snitch, but also understand camera motion based just on visual observations, e.g., motion of the background.
> To conclude, the results on Something-Else, STAR and CATER clearly show that our LRR model can deal with highly varied complex real-world visual inputs and reasoning tasks.
>
> **Nature of surrogate tasks:** We consider tasks like object recognition, tracking and re-identification as low-level tasks. These are computer vision tasks which play a crucial role in solving high-level reasoning tasks, like those found in ACRE, CATER, Something-Else and STAR.
>
> Surrogate tasks that are designed to be as simple as possible, while ensuring that our LRR model predictions are grounded in fine-grained low-level details, as discussed in Section 1 and Section 3.4. In the case of spatio-temporal reasoning tasks, e.g., CATER, Something-Else and STAR, model predictions need to be grounded to object motion, actions and interactions. Therefore, we choose the simplest surrogate tasks possible, which are object detection and tracking. Both surrogate tasks are crucial for good performance. To illustrate this, we perform an ablation on the compositional split of the Something-Else dataset where we include only the detection surrogate task (w/o Tracking),
>
> | Method | Top-1 | Top-5 |
> | -------- | :-------: | :-------: |
> | LRR (w/o Tracking) | 56.4  | 81.5 |
> | LRR (Ours) | 62.0 | 86.3|
>
> The results validate the choice of our surrogate tasks and show that removing any of them would have a negative impact on performance. It is possible that additional surrogate tasks could further improve performance and is an interesting direction of future research. We have improved the clarity of Section 3, in particular Section 3.4 (changes in blue), for an improved explanation of surrogate tasks. We look forward to receiving further feedback to help improve clarity.

---

> ### Author Response · Authors · 2023-11-16
> **Response to Reviewer 5Fo1 [2/2]**
>
> **Overfitting:** Thank you for your insightful observation. There is indeed evidence that our surrogate tasks prevent our LRR model from overfitting on datasets such as CATER. Here we report training and test accuracy of our LRR model trained with and without surrogate tasks (see also Table 3) on the moving camera split of CATER.
>
> | Method | Train | Test |
> | -------- | :-------: | :-------: |
> | LRR (w/o Surrogate tasks)| 99.4  | 62.7 |
> | LRR (Ours) | 89.7 | 80.4 |
>
> Similarly, on the compositional split of the Something-Else dataset (see also Table 2),
>
> | Method | Train | Test |
> | -------- | :-------: | :-------: |
> | LRR (w/o Surrogate tasks) | 92.6  |  50.1 |
> | LRR (Ours) | 87.3 | 62.0 |
>
> The results show that our surrogate tasks clearly prevent overfitting as the model is grounded to the fine-grained low-level details and is thus able to better “understand” the task at hand. We have added these results to Appendix C and we will discuss this in more detail in the final version.
>
> **References**
>
> [1] STAR: A Benchmark for Situated Reasoning in Real-World Videos, Wu et. al. NeurIPS 2021.
>
> [2] Self-Chained Image-Language Model for Video Localization and Question Answering, Yu et. al. arXiv 2023.
>
> [3] Blip-2: Bootstrapping language-image pre-training with frozen image encoders and large language models, Li et. al. arXiv 2023.
>
> [4] Internvideo: General video foundation models via generative and discriminative learning, Wang et. al., arXiv 2023.

---

> > ### Author Response · Authors · 2023-11-20
> > **Evaluation Datasets**
> >
> > **STAR Competition Leaderboard** In addition to the results on the STAR validation set presented above (following the protocol of SeViLA (Yu et. al., 2023)), additionally we also evaluate our LRR model on the STAR challenge leaderboard. The results can be found here: https://eval.ai/web/challenges/challenge-page/1325/leaderboard/3328/Mean, where our LRR model achieves 65.75% overall accuracy and currently **ranked 1st** (as of 20th Nov 2023).
> >
> > This again illustrates the capabilities of our LRR model on *visually complex real-world datasets*.

---

> > > ### Comment · Reviewer_5Fo1 · 2023-11-22
> > > **New results**
> > >
> > > Thanks for providing the new results, and a author response. I've updated my score in view of your response.

---

### Official Review · Reviewer_tQdf · 2023-11-06

**Soundness:** 3 good
**Presentation:** 2 fair
**Contribution:** 3 good
**Rating:** 6
**Confidence:** 4

**Summary:**

This paper proposed a Look, Rember, and Reason (LRR) framework to solve video reasoning task. The structure utilized LM to extract visual information by using surrogate grouding tasks and then integrated the grouding information to arrive at a final answer. The authors propose a two-stream vide encoder to capture sccene structure and object motions to learn low-level skills. Experiments on two sythetic dataset and one real-world datset shows the effectiveness of proposed method on complex spatialtemporal and causal reasoning tasks in videos.

**Strengths:**

1. The proposed LRR structure make use of language models to solve surrage groudning task to benefit final video reasoning task. This structural design better utilize the low-level visual skills and information from videos.

2. This paper conduct experiments on two synthetic datasets and one real-world dataset. The proposed methods achieve competative performance three datasets and outperforms other exsisting baseline models, showing the effectiveness of proposed method.

3. Table 2 and 3 also show the performance of LRR without surrogate tasks and two-stream encoder. The ablation study shows the significance of the two proposed components for the overall structure.

**Weaknesses:**

1. Writing, section 3 and Figure 2 is a little unclear and hard to follow.
2. For different surrogate tasks, where do the ground-truth answers such as localization or box come from?

**Questions:**

See section weakness

---

> ### Author Response · Authors · 2023-11-16
> **Response to Reviewer tQdf**
>
> We thank the reviewer for the helpful comments and for recognizing that our LRR model achieves competitive performance on three -- two synthetic datasets and one real-world -- datasets and that it outperforms other existing baseline models.
>
>
> **Writing, section 3 and Figure 2 is a little unclear and hard to follow:** We apologize for any issues with the clarity of the text in Section 3 and Figure 2. We have made extensive changes to Section 1 and 3 to improve clarity of the text, which we highlight in blue. In Section 1 we have improved the explanation of low-level surrogate tasks and, the “remember” and “reason” steps of our LRR framework. In Section 3, we have improved the overview paragraph of LRR model, focusing on the role of surrogate tasks in our LRR framework. In Section 3.1, we have provided more details of the inputs to and outputs from our LRR model. In Section 3.2, we have clarified more details of our spatial and temporal attention components. In Section 3.3, we have clarified our top-down attention scheme. Finally, in Section 3.4 we have endeavored to define and explain our low-level surrogate tasks more clearly. We will continue to improve the manuscript and we would be glad to receive further and more specific feedback about parts of the text and figures to help improve it further for the final version.
>
>
> **For different surrogate tasks, where do the ground-truth answers such as localization or box come from?** We employ the ground-truth annotations, such as localization or bounding boxes, provided by the corresponding datasets, to train our LRR model end-to-end. As shown in Fig. 1 and Fig. 4, the Something-Else dataset provides bounding box annotations and tracks of hands and objects involved in the action (c.f. Section 4 in Materzynska et. al. 2020). These annotations are also used by the state of the art STIN + OIE + NL model (see Table 2). Similarly, for the ACRE dataset, the object class annotations and the state of the blicket machine are provided, and for the CATER dataset the tracks of the objects in the scene are provided. These annotations are also used by the state of art ALOE model (Ding et al. 2021; see Tables 1 and 3). Thus, we only use annotations provided with the target datasets, and these annotations are also used by the prior state of the art models, ensuring a fair comparison. In case such annotations are not available, due to the recent advances in perception models reaching human-level performance it is possible to use off-the-shelf vision models (Ren et al., 2017; Tang et al., 2017; Ye et al., 2022) as discussed in Section 3.4. It is worth noting that the annotations need to be used only at training time and not at test time, so the computational efficiency of models generating the annotations is not critical for any real-world use. Finally, in the case of STAR (see Appendix B in the updated paper PDF and response to all reviewers), we observe that training jointly on multiple tasks, some of which do and some of which don’t provide annotations, the model is able to acquire the required low-level visual skills from those tasks that do provide them and transfer them to those that don’t. This allows the jointly trained model to reach very high accuracy on the scarcely annotated STAR dataset.

---

> > ### Comment · Reviewer_tQdf · 2023-11-22
> >
> > I'd like to thank the authors for the response and revision of the paper. I think these responses basically address my concerns. I'll make the decision on my final scores after reading other rebuttal materials and discussing with other reviewers

---

### Author Response · Authors · 2023-11-16
**Response to all Reviewers**

We thank all reviewers for the helpful comments. We thank the reviewers for recognizing that, our LRR model “achieves competitive performance on three -- two synthetic datasets and one real-world – datasets” (Reviewer tQdf), demonstrates strong performance on multiple challenging visual reasoning datasets (Reviewer 5Fo1), “framework is innovative” (Reviewer 4932) and that our proposed surrogate tasks such as “object detection, re-identification, and tracking, endows our LRR model with the required low-level visual capabilities” (Reviewer Fa1W).

Here we would like to address common concerns of the reviewers and highlight additional results that are instrumental in addressing the same.

**Visual complexity of evaluation datasets:** We would like to emphasize that recent work has shown that even on synthetic datasets such as ACRE, state of the art LLMs such as GPT-4 do not perform well (Gendron et. al. 2023). Moreover, we include results on the challenging real-world Something-Else dataset in Section 4.2. We additionally include results on the challenging real-world STAR (Wu et. al. 2021) dataset below and in Appendix B in the paper.

The STAR benchmark (https://bobbywu.com/STAR/) consists of questions built upon real-world videos associated with human actions and interactions. Our LRR model with an OPT-350M LM backbone **achieves 66.7% accuracy** and outperforms prior state of the art models such as SeViLA (Yu et. al., 2023), Internvideo (Wang et.al. 2022) and BLIP-2 (Li et. al. 2023). Further details can be found in Appendix B, where we also include ablations of our LRR model (w/o Surrogate tasks) which highlights the advantage of using surrogate tasks during training.

| Method | Int. | Seq. | Pre. | Fea. | Overall|
| -------- | :-------: | :-------: | :-------: | :-------: | :-------: |
| Internvideo | 62.7 | 65.6 | 54.9 | 51.9 | 58.7 |
| BLIP-2 | 65.4 | 69.0 | 59.7 | 54.2 | 62.0 |
| SeViLA | 63.7 | 70.4 | 63.1 | **62.4** | 64.9 |
| LRR (ours) | **79.6** | **71.3** | **63.2** | 52.8 | **66.7** |

Note that methods such as SeViLA are trained on a much larger set of image/video - text pairs and, use the 4.1B parameter BLIP as a video encoder and the 3B parameter Flan-T5 XL as an language model (c.f. Section 4.4 in Yu et al. (2023)). In contrast, our LRR model is trained on a much smaller dataset but one that includes low-level surrogate tasks to endow the model with the requisite low-level visual capabilities.

**Writing and clarity:** We apologize for any issues with the clarity of the text in Section 3 and Figure 2. We have made extensive changes to Section 1 and 3 to improve clarity of the text, which we highlight in blue. In Section 1 we have improved the explanation of low-level surrogate tasks and, the “remember” and “reason” steps of our LRR framework. In Section 3, we have improved the overview paragraph of LRR model, focusing on the role of surrogate tasks in our LRR framework. In Section 3.1, we have provided more details of the inputs to and outputs from our LRR model. In Section 3.2, we have clarified more details of our spatial and temporal attention components. In Section 3.3, we have clarified our top-down attention scheme. Finally, in Section 3.4 we have endeavored to define and explain our low-level surrogate tasks more clearly. We would be glad to receive further and more specific feedback about parts of the text to help improve it further for the final version.

Finally, we look forward to further discussion.

---

> ### Author Response · Authors · 2023-11-20
> **Evaluation Datasets**
>
> **STAR Competition Leaderboard** In addition to the results on the STAR validation set presented above (following the protocol of SeViLA (Yu et. al., 2023)), additionally we also evaluate our LRR model on the STAR challenge leaderboard. The results can be found here: https://eval.ai/web/challenges/challenge-page/1325/leaderboard/3328/Mean, where our LRR model achieves 65.75% overall accuracy and currently **ranked 1st** (as of 20th Nov 2023).
>
> This again illustrates the capabilities of our LRR model on *complex real-world datasets*.

---

### Author Response · Authors · 2023-11-22
**Request for Discussion**

Dear Reviewers,

We would like to thank you for your helpful feedback which has helped us improve the paper.

We addressed reviewers concerns in the author responses, posted on the 16th of Nov 2023. We would be really glad if you could please take a look at our detailed responses so that we can address any remaining concerns before the end of the discussion phase.


Sincerely,

Authors of Submission 3946

---

### Meta-Review · Program_Chairs · 2023-12-11

**Metareview:**

Four experts reviewed this paper with all accepted recommendations. The area chairs are in consensus that this study significantly contributes to training LLM end-to-end on low-level surrogate tasks. These tasks encompass object detection, re-identification, and tracking, ultimately equipping the model with essential low-level visual capabilities. Reviewers did raise some valuable concerns that should be addressed in the final camera-ready version of the paper. The authors are encouraged to make the necessary changes and include the missing references in the final version.

**Justification For Why Not Higher Score:**

The experiments are limited. It would be great to show results on either more challenging synthetic video reasoning datasets (e.g. CLEVRER and ComPhy)or more real-world video reasoning datasets.

**Justification For Why Not Lower Score:**

Interesting idea with solid results.

---

### Decision · Program_Chairs · 2024-01-16

Accept (poster)